

# How a skyrmion can appear both massive and massless

Xiaofan Wu and Oleg Tchernyshyov⋆

William H. Miller III Department of Physics and Astronomy,
Johns Hopkins University, Baltimore, MD 21218, USA

⋆ olegt@jhu.edu

## Abstract

When a magnetic skyrmion is modeled as a point particle, its dynamics depends on the precise definition of the skyrmion center. The guiding-center position, defined as the first moment of the skyrmion density, exhibits Thiele's massless dynamics; position based on the first moment of magnetization component $m_z$ shows Larmor oscillations characteristic of a massive particle. We show that, even with the latter definition, the Larmor oscillations may be absent for certain types of external forces such as adiabatic spin torque. We offer an alternative mechanical model of a skyrmion featuring two coupled massless particles.



## Contents

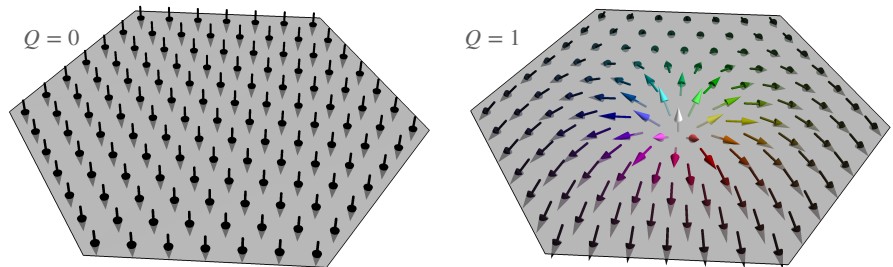

Figure 1: A uniform state ($Q = 0$) and a Belavin-Polyakov [9] skyrmion ($Q = 1$).

# 1 Introduction

Topological solitons in magnets attract the interest of both physicists and engineers [1]. Their stability on the one hand and mobility on the other make them attractive for storing and processing information. Current experimental efforts are focused on domain walls [2,3] and skyrmions [4–7], previously known as magnetic bubbles [8]. A thorough understanding of the dynamics of topological solitons in magnets is a prerequisite for the success of these efforts.

## 1.1 Skyrmion

A skyrmion in a two-dimensional ferromagnet with continuous spatial coordinates $\mathbf{r} = (x, y, 0)$ is a soliton with nontrivial topology of the magnetization field $\mathbf{m} = (m_x, m_y, m_z)$ normalized for convenience to a unit length $|\mathbf{m}| = 1$. A smooth magnetization field $\mathbf{m}(\mathbf{r})$ approaching a uniform state at spatial infinity is characterized by an integer topological invariant defined as the degree of mapping $\mathbf{r} \mapsto \mathbf{m}$,

$$Q = \frac{1}{4\pi} \int dx \, dy \, \mathbf{m} \cdot (\partial_x \mathbf{m} \times \partial_y \mathbf{m}) . \tag{1}$$

States with different skyrmion numbers $Q$ (Fig. 1) cannot be continuously deformed into one another so long as the boundary condition $\mathbf{m} \rightarrow \text{const}$ as $\mathbf{r} \rightarrow \infty$ is maintained.

Skyrmions—stable isolated solitons with $Q = \pm 1$—exist in a number of ferromagnetic models, including the pure, SO(3)-symmetric Heisenberg model [9] as well as its anisotropic variations with additional interactions: chiral Dzyaloshinskii-Moriya terms [10] and long-range dipolar forces [11].

Setting aside the (rather complex) question of skyrmion energetics, we assume that an isolated $Q = 1$ skyrmion in equilibrium is centered at the origin and has a round, axially symmetric shape, Fig. 1. Under these assumptions, it can be parametrized as follows:

$$\theta(\mathbf{r}) = \Theta(r) , \quad \Theta(0) = 0 , \quad \Theta(\infty) = \pi , \qquad \phi(\mathbf{r}) = \alpha + \text{const} . \tag{2}$$

Here $(r, \alpha)$ are polar coordinates and $(\theta, \phi)$ are spherical angles,

$$x + iy = re^{i\alpha} , \quad m_x + im_y = \sin\theta \, e^{i\phi} , \quad m_z = \cos\theta . \tag{3}$$

## 1.2 Skyrmion dynamics

The dynamics of a magnetization field $\mathbf{m}(\mathbf{r}, t)$ is described by the Landau-Lifshitz equation,

$$\mathcal{S}\partial_t \mathbf{m} = -\mathbf{m} \times \frac{\delta U}{\delta \mathbf{m}} , \tag{4}$$

where $\mathcal{S}\mathbf{m}(\mathbf{r})$ is the local density of angular momentum, $U[\mathbf{m}(\mathbf{r})]$ is the energy functional, and $\delta U/\delta\mathbf{m}(\mathbf{r})$ is its functional derivative [12]. (We neglect the damping effects.) Eq. (4) is a nonlinear partial differential equation with precious few exact solutions.

A general method to find an approximate solution for a moving ferromagnetic soliton was suggested by Thiele [13]. Under the assumption that a moving soliton preserves its shape, we may express the magnetization field $\mathbf{m}(\mathbf{r}, t)$ at an arbitrary time $t$ by rigidly translating its initial configuration $\mathbf{m}_0(\mathbf{r}) \equiv \mathbf{m}(\mathbf{r}, 0)$

$$\mathbf{m}(\mathbf{r}, t) = \mathbf{m}_0(\mathbf{r} - \mathbf{R}(t)). \tag{5}$$

Time evolution of the soliton displacement $\mathbf{R}(t) = (X, Y, 0)$ is given by the Thiele equation,

$$\mathbf{G} \times \dot{\mathbf{R}} + \mathbf{F} = 0, \tag{6}$$

expressing the balance of forces acting on the soliton. The first term represents a gyroscopic force linked to the precessional motion of spins. The gyrovector $\mathbf{G} = (0, 0, G)$ is perpendicular to the film plane; its magnitude $G = 4\pi Q\mathcal{S}$ is proportional to the skyrmion number (1) and the spin density $\mathcal{S}$. The second term $\mathbf{F}$ in the Thiele equation (6) represents all other forces acting on the soliton. For instance, position dependence of the soliton energy $U(\mathbf{R})$ creates a conservative force $\mathbf{F} = -\partial U/\partial\mathbf{R}$.

Numerical studies of skyrmion dynamics have uncovered significant deviations from Thiele's theory under fairly mild driving forces [14–19]. The failure can be traced to the breakdown of the rigidity assumption (5). Deformation of a soliton makes the very definition of its position $\mathbf{R}$ ambiguous. Different conventions for $\mathbf{R}$ can yield dramatically different trajectories [15, 19].

### 1.3 Position defined by skyrmion density

Papanicolaou and collaborators [20, 21] defined the skyrmion position $\mathbf{R}$ as the first moment of the skyrmion density $\rho = \frac{1}{4\pi}\mathbf{m} \cdot (\partial_x\mathbf{m} \times \partial_y\mathbf{m})$:

$$\mathbf{R}_{\mathrm{sky}} = \frac{\int dx\,dy\,\rho\,\mathbf{r}}{\int dx\,dy\,\rho}. \tag{7}$$

Trajectory $\mathbf{R}_{\mathrm{sky}}(t)$ shows excellent agreement with Thiele's equation. For example, when a non-uniform magnetic field is suddenly turned on, exerting a constant Zeeman force $\mathbf{F}$ on the skyrmion, the soliton moves in a straight line with a constant velocity perpendicular to the applied force.

It is not a coincidence that $\mathbf{R}_{\mathrm{sky}}(t)$ satisfies the Thiele equation (6) even when the rigidity assumption is violated. $\mathbf{R}_{\mathrm{sky}}$ is directly related to the linear momentum of a skyrmion in two dimensions by the identity [20, 22]

$$\mathbf{P} = -\mathbf{G} \times \mathbf{R}_{\mathrm{sky}}, \tag{8}$$

which is valid for arbitrary deformations of the soliton. When a skyrmion is driven by an external force $\mathbf{F}$, its linear momentum changes at the rate $\dot{\mathbf{P}} = \mathbf{F}$. It follows immediately that $\mathbf{R}_{\mathrm{sky}}(t)$ satisfies the Thiele equation.

### 1.4 Position defined by out-of-plane magnetization

Another common definition of the skyrmion position [14–19, 23] uses the out-of-plane magnetization $m_z$ (relative to its ground-state value of $m_z = -1$) as the statistical weight:

$$\mathbf{R}_{\mathrm{mag}} = \frac{\int dx\,dy\,(m_z + 1)\mathbf{r}}{\int dx\,dy\,(m_z + 1)}. \tag{9}$$

For example, magnetic dichroism of X-rays allows the mapping of the magnetization component $m_z(\mathbf{r}, t)$ with sufficient spatial and temporal resolutions to determine the skyrmion trajectory [24].

In numerical simulations, $\mathbf{R}_{\mathrm{mag}}(t)$ shows marked deviations from Thiele's equation [14–17]. A suddenly switched on Zeeman force generates a cycloidal trajectory, a superposition of the linear motion and a cyclotron orbit [19]. An excellent description of this trajectory is obtained by endowing the Thiele equation with an inertial term [14, 15, 17],

$$m\ddot{\mathbf{R}} = \mathbf{G} \times \dot{\mathbf{R}} + \mathbf{F}. \tag{10}$$

The empirically introduced skyrmion mass $m$ determines the frequency of cyclotron motion $\boldsymbol{\omega} = \mathbf{G}/m$. The origin of this mass can be traced to the deformation of the skyrmion that increases linearly with the velocity of the driven skyrmion. The energy cost is quadratic in the deformation and thus in the velocity, effectively producing a kinetic energy $m|\dot{\mathbf{R}}|^2/2$. Emergence of inertia in ferromagnetic domain walls in the form of kinetic energy was first proposed in theory by Döring [25] and has been observed experimentally [26, 27]. Ivanov and Stephanovich applied this line of argument to skyrmions [28].

## 1.5 Controversy over the "right" definition

Is there a compelling reason to prefer one definition of the skyrmion position over another? Kravchuk *et al.* [29] have recently argued in favor of $\mathbf{R}_{\mathrm{sky}}$, going as far as to declare $\mathbf{R}_{\mathrm{mag}}$ "not physically sound because it does not describe the skyrmion displacement in the sense of the traveling-wave model." We are not convinced by this argument. The assumption of rigid displacement has no profound principle behind it. It was adopted by Thiele for a practical task of describing steady-state motion of a soliton. A closing remark in his letter [13] clearly states that this approach "may be used as a first approximation" beyond steady-state situations. We thus see no fundamental reason why $\mathbf{R}_{\mathrm{sky}}$ should be prefrable to $\mathbf{R}_{\mathrm{mag}}$.

In our view, both definitions can be useful. $\mathbf{R}_{\mathrm{sky}}$ is clearly more convenient because of the simplicity of its dynamics. However, it is $\mathbf{R}_{\mathrm{mag}}$ that is experimentally accessible at the moment [24]. This situation is entirely analogous to the dynamics of a massive charged particle in the presence of a magnetic field. The physical position of the particle—the analog of $\mathbf{R}_{\mathrm{mag}}$—exhibits cyclotron oscillations on top of a steady drift in the direction orthogonal to an external force. The guiding center of the cyclotron orbit—-the analog of $\mathbf{R}_{\mathrm{sky}}$—exhibits a simpler motion without the oscillations. The guiding center is a useful tool in the analysis of low-energy physics of the $n = 0$ Landau level (such as the quantum Hall effects). However, it cannot replace the physical position of the electron if we want to include the degrees of freedom beyond the lowest Landau level.

## 1.6 Outline of the paper

In the remainder of this paper, we intend to show that the presence or absence of Larmor oscillations does not boil down to the subjective choice of the skyrmion coordinates. Even if we use the oscillation-prone $\mathbf{R}_{\mathrm{mag}}$ to describe the skyrmion position, there are realistic physical situations in which the oscillations disappear and the skyrmion motion is described by the inertia-free Thiele equation. An early example of that was found in a numerical study of Schütte *et al.* [18]. The goal of our paper is to elucidate the split personality of the skyrmion through a simple analytical model.

To see how such an outcome is possible, recall that the skyrmion's kinetic energy is simply the potential energy of its deformation in disguise. If the driving force avoids deforming the skyrmion then there will be no kinetic energy and no Larmor oscillations. Put differently, reducing the skyrmion description to just its position is a coarse-graining procedure that discards

its numerous hard modes. Most of these modes can be safely integrated out without affecting the dynamics of translational motion. However, one hard mode is special because it serves as the canonical momentum for the remaining mode $\mathbf{R}_{\mathrm{mag}}$. Integrating out the canonical momentum generates kinetic energy for skyrmion translations. If this hard mode also couples to the driving force then there is an additional effect: integrating out the hard mode also modifies the coupling of $\mathbf{R}_{\mathrm{mag}}$ to the driving force. Under the right circumstances, the modified driving force will not cause any Larmor oscillations.

More detailed explanations are provided in the following sections. In Sec. 2 we discuss two equivalent models for translational motion of a skyrmion. One of them is the familiar model of a massive particle in a magnetic field equivalent to the modified Thiele equation with a mass term (10); the other has a massless particle coupled to an invisible partner via potential and gyroscopic forces. Integrating out the invisible particle generates a mass and modifies the driving force. In Sec. 3 we show that the second model describes the modes of a skyrmion bubble relevant to its translational motion and give a physical example of the driving force—spin-transfer torque—that creates no Larmor oscillations. Sec. 4 contains concluding remarks.

## 2 Models of Skyrmion Translational Motion

### 2.1 Massive particle in a magnetic field

We begin with the familiar example of a massive particle moving in a uniform magnetic field. The particle is confined to the $xy$ plane, so that $\mathbf{r} = (x, y, 0)$, and the field is normal to the plane, $\mathbf{B} = (0, 0, B)$. We add an electric field $\mathbf{E} = (E, 0, 0)$ as the driving force.

The equation of motion for the particle is

$$m\ddot{\mathbf{r}} = \frac{e}{c}\dot{\mathbf{r}} \times \mathbf{B} + e\mathbf{E}. \tag{11}$$

Eq. (11) is an inhomogeneous linear differential equation for $\mathbf{r}$. It has a partial solution in the form of uniform motion

$$\mathbf{r}(t) = \mathbf{r}(0) + \mathbf{v}_d t, \quad \mathbf{v}_d = \frac{\mathbf{E} \times \mathbf{B}}{B^2}c, \tag{12}$$

where the drift velocity $\mathbf{v}_d$ is perpendicular to the direction of the electric field. The general solution of Eq. (11) is a superposition of drift (12) and circular motion at the Larmor angular frequency

$$\boldsymbol{\omega} = -\frac{e\mathbf{B}}{mc}. \tag{13}$$

It can be seen that the drift (12) is a purely gyroscopic effect independent of inertia. In contrast, the Larmor frequency (13) is explicitly dependent on mass $m$; therefore, Larmor rotation reveals inertia.

Whether the inertial effects are seen in the particle's dynamics depends on the precise way in which it is set in motion. With the particle initially at rest at the origin, switching on the electric field suddenly at $t = 0$ puts the particle on a cycloidal trajectory with equal speeds of drift and rotational motion:

$$\mathbf{v}(t) = \frac{Ec}{B}(\sin \omega t, \cos \omega t - 1, 0), \quad \mathbf{r}(t) = \frac{Ec}{B\omega}(1 - \cos \omega t, \sin \omega t - \omega t, 0). \tag{14}$$

If, on the other hand, the electric field $\mathbf{E}(t)$ is turned on gradually, so that it does not change appreciably during a Larmor period $T = 2\pi/\omega$, the Larmor rotation will be absent and the particle will always be moving at the instantaneous drift velocity (12).

The equation of motion (11) can be rewritten as $\dot{\mathbf{P}} = e\mathbf{E}$, where

$$\mathbf{P} = m\dot{\mathbf{r}} - \frac{e}{c}\mathbf{r} \times \mathbf{B} \tag{15}$$

is a generalized version of linear momentum in a uniform magnetic field. In the absence of the external force $e\mathbf{E}$ it is conserved, hence the name "conserved linear momentum." In quantum mechanics $\mathbf{P}$ is known as the generator of "magnetic translations" [30].

Conserved linear momentum can be expressed geometrically as a position known as the guiding center of the cyclotron orbit $\mathbf{r}_{\mathrm{gc}}$:

$$\mathbf{P} = -\frac{e}{c}\mathbf{r}_{\mathrm{gc}} \times \mathbf{B}, \tag{16}$$

where

$$\mathbf{r}_{\mathrm{gc}} = \mathbf{r} - \frac{\boldsymbol{\omega} \times \dot{\mathbf{r}}}{\omega^2}. \tag{17}$$

When the external force is absent and the particle moves in a circular orbit, $\mathbf{r}_{\mathrm{gc}}$ is conserved. Under an external force, the guiding center drifts sideways:

$$\frac{e}{c}\dot{\mathbf{r}}_{\mathrm{gc}} \times \mathbf{B} + e\mathbf{E} = 0. \tag{18}$$

The guiding center moves without inertia, just like the skyrmion coordinate $\mathbf{R}_{\mathrm{sky}}$.

## 2.2 Massless particle with an invisible partner

Inertia is not a property inherent to spins. Their dynamics is purely precessional. Inertia in the context of a ferromagnet is an emergent property. When a soft mode of a ferromagnet is coupled gyroscopically to a hard mode, integrating out the hard mode creates effective kinetic energy for the soft mode. The model of a massive particle considered in Sec. 2.1 can also be derived from a basic model of two particles with no inertia.

Consider two massless particles at positions $\mathbf{r}_1$ and $\mathbf{r}_2$ confined to move in the $xy$ plane. They are coupled by a spring force that compels them to move more or less together. In addition, the particles are coupled by a Lorentz-like force that is somewhat unusual: the force on particle 1 is proportional to the velocity of particle 2 and vice versa. Finally, there is an external electric field to which they couple with different strengths determined by their electric charges. The equations of motion are

$$\begin{aligned} \text{Particle 1:} \quad & \frac{q}{c}\dot{\mathbf{r}}_2 \times \mathbf{B} + k(\mathbf{r}_2 - \mathbf{r}_1) + q_1\mathbf{E} = 0, \\ \text{Particle 2:} \quad & \frac{q}{c}\dot{\mathbf{r}}_1 \times \mathbf{B} + k(\mathbf{r}_1 - \mathbf{r}_2) + q_2\mathbf{E} = 0. \end{aligned} \tag{19}$$

Note that the coupling to the electric field is defined by the electric charges $q_1$ and $q_2$, which are distinct from the coupling $q$ for the mutual magnetic field $\mathbf{B}$. Furthermore, because $q$ and $\mathbf{B}$ appear only as a product, we are free to rescale them as long as the product $q\mathbf{B}$ stays unchanged. It will be convenient to set $2q$ equal to the net electric charge:

$$q_1 + q_2 = 2q. \tag{20}$$

Eqs. (19) can be simplified by the introduction of normal modes, position of the guiding center $\mathbf{r}_{\mathrm{gc}} = (\mathbf{r}_1 + \mathbf{r}_2)/2$ and relative position $\mathbf{r}_{\mathrm{rel}} = \mathbf{r}_1 - \mathbf{r}_2$:

$$\begin{aligned} \text{Guiding center:} \quad & \frac{2q}{c}\dot{\mathbf{r}}_{\mathrm{gc}} \times \mathbf{B} + 2q\mathbf{E} = 0, \\ \text{Relative motion:} \quad & -\frac{q}{2c}\dot{\mathbf{r}}_{\mathrm{rel}} \times \mathbf{B} - k\mathbf{r}_{\mathrm{rel}} + \frac{q_1 - q_2}{2}\mathbf{E} = 0. \end{aligned} \tag{21}$$

The guiding center moves in the direction orthogonal to the electric field with the drift velocity (12). The relative motion is rotation at the Larmor frequency (13) with $e = 2q$.

If the observer can only see the center position $\mathbf{r}_{\mathrm{gc}}$ then he or she will find that the motion of the center exhibits no inertial effects. Suppose, however, that the observer can only see particle 1 but not particle 2. Then we should eliminate $\mathbf{r}_2$ from the equations of motion and express everything in terms of $\mathbf{r}_1$. The resulting dynamics is reminiscent of a massive particle (11):

$$m\ddot{\mathbf{r}}_1 = \frac{2q}{c}\dot{\mathbf{r}}_1 \times \mathbf{B} + 2q\mathbf{E} + \frac{q_2}{q}\frac{\dot{\mathbf{E}} \times \mathbf{B}}{B^2}mc. \tag{22}$$

The effect of integrating out $\mathbf{r}_2$ is threefold. First, particle 1 acquires a mass

$$m = \frac{q^2 B^2}{kc^2}. \tag{23}$$

Second, its electric charge is renormalized from $q_1$ to $q_1 + q_2 = 2q$, indicating that the external force $q_2\mathbf{E}$, formerly applied to particle 2, has been transferred to particle 1. Third, a new dynamical force, proportional to $\dot{\mathbf{E}} \times \mathbf{B}$, arises if the electric field varies in time.

It is instructive to examine what happens when we switch on the electric field suddenly at $t = 0$, with the particles initially at rest, $\mathbf{r}_1 = \mathbf{r}_2 = 0$. Because of the extra force, particle 1 receives a kick instantaneously increasing its velocity to

$$\dot{\mathbf{r}}_1(+0) = \frac{q_2}{q}\frac{\mathbf{E} \times \mathbf{B}}{B^2}c. \tag{24}$$

If the particles have the same electric charge, $q_1 = q_2 = q$, then this velocity exactly equals the drift velocity (12). In this case, particle 1 will keep moving at the drift velocity without Larmor oscillations.

Although the exact cancellation of Larmor oscillations at $q_1 = q_2$ looks like a happy coincidence, there is, in fact, a deeper principle at work. When $q_1 = q_2$, Eq. (21) tells us that the normal mode $\mathbf{r}_{\mathrm{rel}}$, responsible for Larmor oscillations, has no coupling to the electric field. It does not get excited and therefore $\mathbf{r}_1 = \mathbf{r}_{\mathrm{gc}} + \mathbf{r}_{\mathrm{rel}}/2 = \mathbf{r}_{\mathrm{gc}}$. Even though the observer is watching particle 1, its motion is the same as that of the guiding center, so it does not exhibit inertia.

This example demonstrates that the dynamics of a system with emergent inertia may or may not exhibit inertial effects such as Larmor oscillations if the external force that sets the system in motion couples to it in a particular way.

Equations of motion (19) can be obtained from the following Lagrangian:

$$L(\mathbf{r}_1, \mathbf{r}_2) = -\frac{q}{c}\mathbf{B} \cdot (\dot{\mathbf{r}}_1 \times \mathbf{r}_2) - \frac{k(\mathbf{r}_1 - \mathbf{r}_2)^2}{2} + (q_1\mathbf{r}_1 + q_2\mathbf{r}_2) \cdot \mathbf{E}. \tag{25}$$

The Lagrangian can also be expressed in terms of the guiding center and relative coordinate:

$$L(\mathbf{r}_{\mathrm{gc}}, \mathbf{r}_{\mathrm{rel}}) = -\frac{q}{c}\mathbf{B} \cdot (\dot{\mathbf{r}}_{\mathrm{gc}} \times \mathbf{r}_{\mathrm{gc}}) + 2q\mathbf{r}_{\mathrm{gc}} \cdot \mathbf{E} + \frac{q}{4c}\mathbf{B} \cdot (\dot{\mathbf{r}}_{\mathrm{rel}} \times \mathbf{r}_{\mathrm{rel}}) - \frac{kr_{\mathrm{rel}}^2}{2} + \frac{q_1 - q_2}{2}\mathbf{r}_{\mathrm{rel}} \cdot \mathbf{E}. \tag{26}$$

It is evident from this Lagrangian that the electric field is decoupled from relative motion when $q_1 = q_2$, so Larmor oscillations are not induced in this case.

## 3 Translational Motion of a Skyrmion Bubble

### 3.1 Heuristic argument

A quick way to see that the two-particle model described in Sec. 2.2 applies to skyrmion dynamics is through a modest generalization of Thiele's approach. Instead of allowing rigid

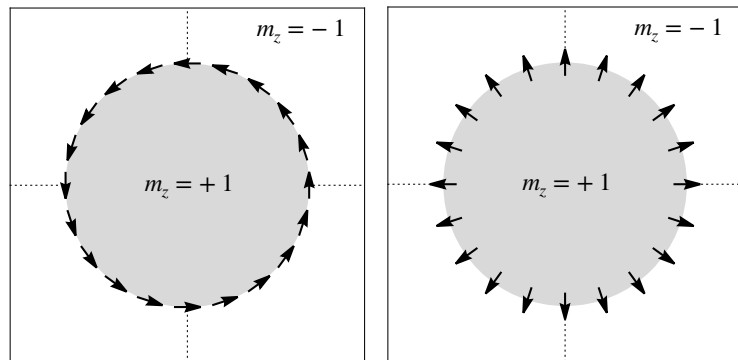

Figure 2: A $Q = +1$ skyrmion bubble in equilibrium. Arrows show the direction of the in-plane magnetization components $(m_x, m_y)$ on a Bloch (left) or Neél (right) domain wall separating domains with $m_z = +1$ (gray) and $m_z = -1$ (white).

translations of the entire magnetization field $\mathbf{m}(\mathbf{r}) \mapsto \mathbf{m}(\mathbf{r} - \mathbf{R})$, we do the same with the fields of spherical angles $\theta$ and $\phi$ and let them shift independently from each other:

$$\theta(\mathbf{r}) \mapsto \theta(\mathbf{r} - \mathbf{r}_1), \quad \phi(\mathbf{r}) \mapsto \phi(\mathbf{r} - \mathbf{r}_2). \tag{27}$$

This yields two pairs of collective coordinates $\mathbf{r}_1$ and $\mathbf{r}_2$ instead of just one in Thiele's method. The fields $\theta$ and $\phi$ are of course coupled: we do not expect the center of $\theta(\mathbf{r} - \mathbf{r}_1)$ to run away from the center of $\phi(\mathbf{r} - \mathbf{r}_2)$. We may therefore anticipate the presence of a rotationally symmetric parabolic confining potential $U = k(\mathbf{r}_1 - \mathbf{r}_2)^2/2$. In addition, there is a gyroscopic coupling, as we shall see next.

Equations of motion for general collective coordinates $q = \{q^1, q^2, \ldots\}$ of a ferromagnetic soliton are [31,32]

$$F_{ij}\dot{q}^j - \frac{\partial U}{\partial q^i} = 0. \tag{28}$$

Here we omitted the effects of viscous friction; $F_{ij}$ is the antisymmetric gyroscopic tensor

$$F_{ij} = -F_{ji} = -\mathcal{S}\int dV\, \mathbf{m} \cdot \left(\frac{\partial \mathbf{m}}{\partial q^i} \times \frac{\partial \mathbf{m}}{\partial q^j}\right) = -\mathcal{S}\int dV \sin\theta \left(\frac{\partial \theta}{\partial q^i}\frac{\partial \phi}{\partial q^j} - \frac{\partial \theta}{\partial q^j}\frac{\partial \phi}{\partial q^i}\right), \tag{29}$$

and $\mathcal{S}$ is the spin density in a fully magnetized state. We assume an axially symmetric skyrmion (2) and use the identities

$$\frac{\partial \theta}{\partial x_1} = -\frac{\partial \theta}{\partial x} = -\frac{d\theta}{dr}\cos\alpha, \quad \frac{\partial \phi}{\partial y_2} = -\frac{\partial \phi}{\partial y} = -\frac{1}{r}\frac{d\phi}{d\alpha}\cos\alpha, \tag{30}$$

and set $\mathbf{r}_1 = \mathbf{r}_2 = 0$ to obtain

$$F_{x_1 y_2} = -F_{y_2 x_1} = -\mathcal{S}\int_0^\infty r\,dr \int_0^{2\pi} d\alpha \sin\theta \frac{d\theta}{dr}\frac{1}{r}\frac{d\phi}{d\alpha}\cos^2\alpha = -2\pi\mathcal{S}. \tag{31}$$

We also find, along similar lines, that $F_{x_2 y_1} = -F_{y_1 x_2} = -2\pi\mathcal{S}$. All other components of the gyroscopic tensor vanish.

Substitution of the gyroscopic tensor and potential energy into Eq. (28) immediately yields the equations of motion of the two-particle model (19) with $\frac{q}{c}\mathbf{B} = (0, 0, -2\pi\mathcal{S})$, albeit without an electric field, which requires an external perturbation.

## 3.2 Basic modes of a magnetic skyrmion

We shall now derive the results anticipated in Sec. 3.1 in a more systematic way. We do so for a skyrmion bubble, a type of skyrmion that exists in thin ferromagnetic films with easy-axis anisotropy. A bubble with skyrmion number $Q = +1$ can be described as a circular domain with magnetization $\mathbf{m} = (0, 0, +1)$ separated by a domain wall from the surrounding domain with $\mathbf{m} = (0, 0, -1)$, Fig. 2. The equilibrium radius of the bubble $\bar{r}$ is determined by a competition between the easy-axis anisotropy and long-range dipolar interactions and is typically much larger than the characteristic width of the domain wall.

It is convenient to identify the domain wall with the line on which the easy-axis component of magnetization vanishes, $m_z = 0$. On this line, the magnetization lies in the hard plane $xy$ and is coupled to the direction of the domain wall by the dipolar [17] or Dzyaloshinskii-Moriya interactions [33]. In equilibrium, the in-plane magnetization is typically either parallel to the domain wall (Bloch skyrmion, left panel of Fig. 2) or perpendicular to it (Néel skyrmion, right panel of Fig. 2).

Low-energy states of a circular bubble can be conveniently parametrized in terms of polar coordinates $(r, \alpha)$ in the $xy$ plane: $r(\alpha)$ gives the position of the domain wall and $\phi(\alpha)$ the azimuthal angle of in-plane magnetization. Low-energy dynamics of the bubble is confined to its boundary and can be viewed as slow variations of these fields. It can be obtained from the Lagrangian [17] for the fields $r(t, \alpha)$ and $\phi(t, \alpha)$,

$$L[r(\alpha), \phi(\alpha)] = \bar{r} \int_0^{2\pi} d\alpha \left( -2\mathcal{S} \frac{\partial r}{\partial t} \phi - \frac{\kappa}{2} (\phi - \phi_{\text{eq}})^2 \right) - U[r(\alpha)]. \tag{32}$$

The first term in the integrand represents the gyroscopic force and comes from the spin Berry phase; $\mathcal{S}$ is the spin density per unit area (in a uniform ground state). The second, potential term is the cost of the in-plane magnetization deviating from its equilibrium orientation for a given shape of the boundary $r(\alpha)$,

$$\phi_{\text{eq}}(\alpha) = \alpha + \delta - \frac{1}{\bar{r}} \frac{\partial r}{\partial \alpha}. \tag{33}$$

Here $\delta = \pm \pi/2$ (Bloch domain wall) or $0, \pi$ (Néel domain wall). Lastly, the functional $U[r(\alpha)]$ in Eq. (32) is the part of potential energy that depends on the shape of the bubble $r(\alpha)$; this contribution need not concern us because translational motion of the bubble does not affect its shape.

The fields $r(\alpha)$ and $\phi(\alpha)$ are conveniently expressed in terms of Fourier amplitudes:

$$r(\alpha) = \bar{r} + \sum_{m=0}^{\infty} (a_m \cos m\alpha + b_m \sin m\alpha),$$
$$\phi(\alpha) = \alpha + \delta + \sum_{m=0}^{\infty} (\xi_m \cos m\alpha + \eta_m \sin m\alpha). \tag{34}$$

Amplitude $a_0$ describes the breathing mode of the bubble; $a_2$ and $b_2$ quantify elliptic deformations of its boundary. Displacements of the circular boundary without changes in its size or shape are described by amplitudes $a_1$ and $b_1$ (left panels of Fig. 3). As the boundary is defined as the locus of points where $\cos \theta = 0$, we may identify $a_1$ and $b_1$ as the rigid displacements of the $\theta$ field from Sec. 3.1,

$$\mathbf{r}_1 = (a_1, b_1). \tag{35}$$

A rigid translation of the $\phi$ field,

$$\phi(\mathbf{r}) \mapsto \phi(\mathbf{r} - \mathbf{r}_2) = \phi(\mathbf{r}) - \mathbf{r}_2 \cdot \nabla \phi(\mathbf{r}) = \phi(\mathbf{r}) + \frac{x_2 \sin \alpha - y_2 \cos \alpha}{r}. \tag{36}$$

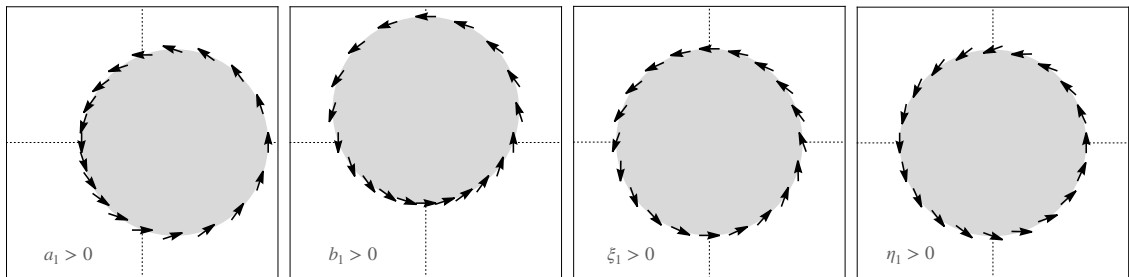

Figure 3: Basic modes of a skyrmion bubble with the azimuthal number $m = 1$. Left panels: $\mathbf{r}_1$. Right panels: $\mathbf{r}_2$.

Comparison with Eq. (34) yields

$$\mathbf{r}_2 = (\bar{r}\eta_1, -\bar{r}\xi_1). \tag{37}$$

These two modes are shown in the right panels of Fig. 3.

Upon substituting the Fourier expansion (34) into the Lagrangian of the domain wall (32), we obtain the Lagrangian of the four $m = 1$ modes:

$$L(a_1, b_1, \xi_1, \eta_1) = -2\pi\bar{r}\mathcal{S}(\dot{a}_1\xi_1 + \dot{b}_1\eta_1) - \frac{\pi\kappa}{2\bar{r}}\left[(a_1 - \bar{r}\eta_1)^2 + (b_1 + \bar{r}\xi_1)^2\right]. \tag{38}$$

Comparison of Eqs. (25) and (38) reveals a precise analogy between the dynamics of a skyrmion bubble and that of two massless particles considered in Sec. 2.2. Thus rigid displacements of the field $\theta$ play the role of the observable particle, whereas those of the field $\phi$ serve as the invisible partner. Integrating out "invisible" $\phi$ amplitudes $\xi_1$ and $\eta_1$ yields a particle of mass $m = 4\pi\bar{r}\mathcal{S}^2/\kappa$, a result derived previously [17].

Note that rigid translations of the bubble correspond to combinations of boundary shifts and twists proportions $a_1 = \bar{r}\eta_1$ and $b_1 = -\bar{r}\xi_1$ (left panels of Fig. 4). These normal modes cost no potential energy and are the analogs of translations in the two-particle model. An orthogonal pair of normal modes with $a_1 = -\bar{r}\eta_1$ and $b_1 = +\bar{r}\xi_1$ (right panels of Fig. 4) correspond to the relative motion. The correspondence is as follows:

$$\mathbf{r}_{\text{gc}} = \frac{1}{2}(a_1 + \bar{r}\eta_1, b_1 - \bar{r}\xi_1, 0), \quad \mathbf{r}_{\text{rel}} = (a_1 - \bar{r}\eta_1, b_1 + \bar{r}\xi_1, 0). \tag{39}$$

Next we consider external forces driving a skyrmion bubble.

### 3.3 Zeeman coupling

The simplest example is provided by the Zeeman coupling to a magnetic field $\mathbf{H} = (0, 0, H)$ applied along the easy $z$-axis [16, 17, 24]. A uniform field exerts uniform pressure on the domain wall of the bubble, which has no effect on the $m = 1$ modes related to displacements.

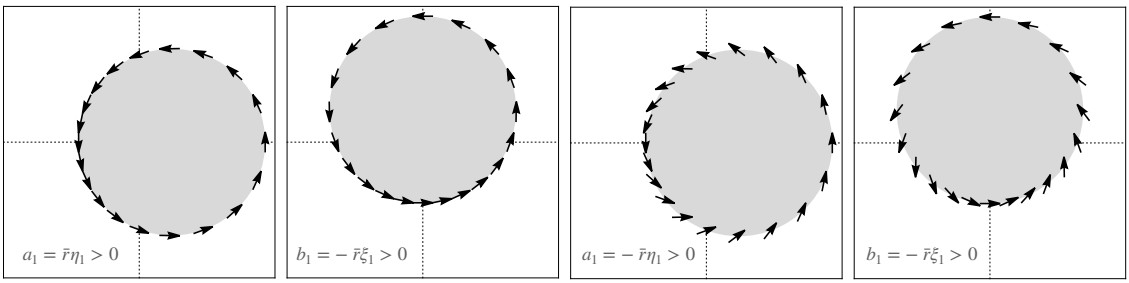

Figure 4: Normal modes of a skyrmion bubble with the azimuthal number $m = 1$. Left panels: $\mathbf{r}_{\text{gc}}$. Right panels: $\mathbf{r}_{\text{rel}}$.

For a field with constant gradient, $H_z(\mathbf{r}) = H_z(0) + \mathbf{r} \cdot \nabla H_z$, the Zeeman coupling endows the displacements with potential energy

$$U(\mathbf{r}_1) = -\gamma \mathcal{S} \int dx\, dy\, H_z(\mathbf{r}) m_z(\mathbf{r}) = U(0) - M_z \mathbf{r}_1 \cdot \nabla H_z\,, \tag{40}$$

where $\gamma$ is the gyromagnetic ratio and $M_z = \gamma \mathcal{S} \int dx\, dy\, (m_z + 1)$ is the magnetic moment of the skyrmion relative to the $m_z = -1$ ground state.

Equations of motion for collective coordinates $\mathbf{r}_1$ and $\mathbf{r}_2$ can now be obtained from the general recipe, Eq. (28):

$$\begin{aligned}
-2\pi \mathcal{S} \dot{\mathbf{r}}_2 \times \mathbf{e}_z + k(\mathbf{r}_2 - \mathbf{r}_1) + M_z \nabla H_z &= 0\,, \\
-2\pi \mathcal{S} \dot{\mathbf{r}}_1 \times \mathbf{e}_z + k(\mathbf{r}_1 - \mathbf{r}_2) &= 0\,.
\end{aligned} \tag{41}$$

Here $\mathbf{e}_z = (0, 0, 1)$ and $k = \pi\kappa/\bar{r}$. We see that the field gradient couples to $\mathbf{r}_1$ but not $\mathbf{r}_2$.

The equations of motion for the guiding center and relative position are

$$\begin{aligned}
-2\pi \mathcal{S} \dot{\mathbf{r}}_{\text{gc}} \times \mathbf{e}_z + \frac{1}{2} M_z \nabla H_z &= 0\,, \\
-\pi \mathcal{S} \dot{\mathbf{r}}_{\text{rel}} \times \mathbf{e}_z - k \mathbf{r}_{\text{rel}} + \frac{1}{2} M_z \nabla H_z &= 0\,.
\end{aligned} \tag{42}$$

The field gradient couples to both the guiding center and relative position. It will therefore not only cause drift but also excite Larmor oscillations, revealing the skyrmion's inertia.

The obtained equations of motion are the same as for two massless particles (19) with the following parameters:

$$q_1 \mathbf{E} = M_z \nabla H_z\,, \quad q_2 \mathbf{E} = 0\,, \quad \frac{q}{c} \mathbf{B} = (0, 0, -2\pi \mathcal{S})\,. \tag{43}$$

## 3.4 Coupling to a spin-polarized electric current

An electric current passing through a ferromagnet becomes spin-polarized along the local direction of magnetization $\mathbf{m}(\mathbf{r})$. Spatial variations of magnetization lead to the rotation of the spins in the flowing current, thereby applying a torque. The current then applies an equal and opposite torque to the magnetization [34–37]. In the adiabatic approximation, the density of the spin-transfer torque is

$$\boldsymbol{\tau}_{\text{st}} = -\frac{P\hbar}{2e} (\mathbf{j} \cdot \nabla) \mathbf{m}\,, \tag{44}$$

where $\mathbf{j}$ is the density of electric current, $e$ is the electron charge, and $P$ is the degree of spin polarization in the current. It is convenient to express the spin torque in terms on an effective velocity $\mathbf{u}$ defined by the relation

$$\mathcal{S} \mathbf{u} = \frac{P\hbar}{2e} \mathbf{j}\,. \tag{45}$$

The Landau-Lifshitz equation (4) then reads

$$\mathcal{S}(\partial_t + \mathbf{u} \cdot \nabla) \mathbf{m} = -\mathbf{m} \times \frac{\delta U}{\delta \mathbf{m}}\,. \tag{46}$$

For a uniform current density $\mathbf{j}$, and hence uniform $\mathbf{u}$, the equations of motion read

$$\begin{aligned}
-2\pi \mathcal{S}(\dot{\mathbf{r}}_2 - \mathbf{u}) \times \mathbf{e}_z + k(\mathbf{r}_2 - \mathbf{r}_1) &= 0\,, \\
-2\pi \mathcal{S}(\dot{\mathbf{r}}_1 - \mathbf{u}) \times \mathbf{e}_z + k(\mathbf{r}_1 - \mathbf{r}_2) &= 0\,.
\end{aligned} \tag{47}$$

The equations of motion for the guiding center and relative position are

$$-2\pi\mathcal{S}(\dot{\mathbf{r}}_{\text{gc}} - \mathbf{u}) \times \mathbf{e}_z = 0\,,$$
$$\pi\mathcal{S}\dot{\mathbf{r}}_{\text{rel}} \times \mathbf{e}_z - k\mathbf{r}_{\text{rel}} = 0\,. \tag{48}$$

It can be seen that the adiabatic spin-transfer torque only couples to the guiding center, but not to the relative motion. Therefore, Larmor oscillations are not excited by an electric current.

Translation to the model with two massless particles is as follows:

$$q_1\mathbf{E} = q_2\mathbf{E} = \frac{q}{c}\mathbf{B} \times \mathbf{u} = -\mathbf{e}_z \times \frac{\pi P\hbar}{e}\mathbf{j}\,. \tag{49}$$

The electric current couples equally strongly to both particles, so $q_1 = q_2 = q$ and the external force produces pure drift with no Larmor oscillations and with an instantaneous velocity proportional to the driving force,

$$\dot{\mathbf{r}}_1 = \dot{\mathbf{r}}_{\text{gc}} + \frac{1}{2}\dot{\mathbf{r}}_{\text{rel}} = \dot{\mathbf{r}}_{\text{gc}} = \mathbf{u}(t)\,. \tag{50}$$

The absence of Larmor oscillations in a skyrmion driven by an electric current was pointed out by Schütte *et al.* [18].

It bears noting that the absence of soliton deformations and its apparently massless dynamics under adiabatic spin-transfer torque are well-known and robust results that transcend the narrow topic of this paper (skyrmions) and its technical method (collective coordinates). That has already been pointed in a general context by Bazaliy *et al.* [34] and for domain walls in one dimension by Barnes and Maekawa [36]. Their line of reasoning was as follows. Suppose a static configuration $\mathbf{m}_0(\mathbf{r})$ minimizes the energy functional $U$ and therefore satisfies the Landau-Lifshitz equation (46) with $\mathbf{u} = 0$. It then follows that the rigid traveling-wave Ansatz (5) is the solution for an arbitrary $\mathbf{u}(t)$, provided that the instantaneous velocity of the wave matches the instantaneous effective velocity, $\dot{\mathbf{R}}(t) = \mathbf{u}(t)$.

This general argument is only applicable for this specific form of external perturbation (adiabatic spin-transfer torque). Other perturbations induce soliton deformations and therefore require the treatment of collective coordinates beyond just simple translations.

## 4 Conclusion

We have presented a simple mechanical model of a skyrmion that resolves a recent controversy about skyrmion mass [19, 29]. The toy model consists of two massless particles coupled to each other by two distinct forces: a parabolic potential and a mutual Lorentz force (19). Only particle 1 is accessible to observations, whereas particle 2 stays hidden.

Loosely speaking, the observable particle in our analogy is associated with the dynamics of the longitudinal magnetization $m_z$ and the hidden particle with that of the transverse ones $m_x$ and $m_y$. We have presented a formal derivation in the limit where the skyrmion is a magnetic bubble—a circular domain of the $m_z = +1$ state separated by a narrow domain wall from a surrounding domain of the $m_z = -1$ state. In this case, particle 1 represents the displacement of the bubble $\mathbf{R}_{\text{mag}}$ (9), whereas particle 2 depicts fluctuations of the transverse magnetization (or its azimuthal angle) on the domain wall. Experiments typically measure $\mathbf{R}_{\text{mag}}$, hence the analogy.

Integrating out invisible particle 2 endows particle 1 with an emergent (Döring) mass, in agreement with our prior result [17]. The emergent inertia manifests itself in Larmor oscillations of relative motion. However, whether these oscillations can be observed is a subtle question. The answer depends on the nature of the driving force, specifically on the relative strengths of its couplings to the two particles. Two idealized cases illustrate this point.

(a) If the external force does not couple at all to the invisible particle then integrating out the hidden degrees of freedom only generates the Döring mass for the observed motion. A sudden onset of this force generates Larmor oscillations. The skyrmion appears massive.

(b) If the external force couples equally strongly to both particles then it is decoupled from the normal mode responsible for Larmor oscillations. The skyrmion appears massless.

It is remarkable that the two limiting cases are realized quite naturally by the two most common external forces available to the experimentalist: (a) the Zeeman energy of an inhomogeneous magnetic field; (b) the adiabatic spin-transfer torque from an electric current. If a generic external perturbation falls somewhere in the middle then the Larmor oscillations will be present, albeit in an attenuated form. So under a generic perturbation a skyrmion will appear massive.

The presence of inertial mass can be traced, as in previous studies [25,28], to a deformation of the moving soliton. Both the velocity of the soliton and its deformation are proportional to the external force driving the motion and, therefore, to one another. The concomitant energy increase is quadratic in the degree of deformation and thus in velocity, which makes it possible to view it as kinetic energy. An important point of this work is that some external perturbations create no deformation and therefore do not generate kinetic energy. Thus the answer to the question of whether the skyrmion is massive or massless depends not just on the properties of the skyrmion itself but also on the nature of the driving force. Our alternative mechanical model of a skyrmion as two coupled massless particles encompasses all such scenarios.

## Acknowledgements

We thank B. A. Ivanov, V. P. Kravchuk, A. Rosch, and D. D. Sheka for valuable discussions.

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
