# Peer review of "How a skyrmion can appear both massive and massless"

_SciPost Physics, doi:SciPost Phys. 12, 159 (2022)_

## Round 3 · Referee Report · Anonymous (Referee 1) · 2022-3-30

Strengths

The paper is written clearly in a good pedagogical manner.

Weaknesses

The paper does not add much to the already existing discussion.

Report

This paper contributes to the discussion about the mass of a ferromagnetic skyrmion. In terms of the collective coordinates, the dynamics of a finite-size skyrmion is reduced to the dynamics of a point -- skyrmion center. It is already well known that the dynamics of the skyrmion center depends on its definition:
(i) If the skyrmion center is defined as the first moment of the topological charge, then the dynamics is massless.
(ii) If the skyrmion center is defined in a manner that implies the coupling of the translational mode with the higher magnon modes, then the dynamics can be massive. For example, the skyrmion center can be defined as the first moment of the $m_z$-component.
The new finding of this paper is that in case (ii) the skyrmion dynamics can be massive or massless depending on the driving force. In addition, the paper is written in a good pedagogical manner with the use of mechanical analogies. Based on this I support publishing in SciPost Physics.
However, I have a number of observations:
1. The abstract is written in a misleading manner. The main statement is that massive or massless behavior depends on the driving force. However, it is not written that this statement refers to case (ii). I suggest a reformulation of the abstract.
2. The meaning of the quantity $G$ in (23) is not explained.
3. I think that the derivation of (35) should be explained in more detail. For me it is not obvious, how (35) follows from (34).

Requested changes

The abstract should be reformulated in a not misleading manner.

  • validity: good
  • significance: ok
  • originality: good
  • clarity: top
  • formatting: excellent
  • grammar: perfect

Author:  Oleg Tchernyshyov  on 2022-04-09  [id 2368]

(in reply to Report 1 on 2022-03-30)

Response to Anonymous Review 1. 1. We have rewritten the abstract to mention the two existing definitions of the skyrmion center, which gives a better framing for our main result. The revised abstract also mentions the alternative mechanical model with two coupled massless particles. 2. Eq. (23) has been corrected: it now contains the magnetic field B instead of the undefined variable G. 3. As can be seen in Fig. 3(a), coordinates $a_1$ and $b_1$ encode a rigid displacement of the skyrmion boundary (defined as the line where $\cos{\theta}=0$). Therefore these modes simply translate the $\theta$ field and leave the $\phi$ field alone. That is explained in the paragraph between Eqs. (34) and (35).

---

## Round 3 · Referee Report · Anonymous (Referee 2) · 2022-4-4

Report

This is a very interesting article, where a fresh idea how to “visualize” dynamics of magnetic skyrmion. Concretely, dynamics of the guiding center (GCD) and Larmor dynamics (LD), naturally associated with two lowest magnon modes upon the skyrmion, are described here by two inertia-free particles. Then two possible regimes of forced motion are found, one for GCD + LD and one for GCD only. This observation is useful step to understanding of the general picture of skyrmion dynamics that is obviously of a great interest for both general magnetics and applications. For me, this article is warrant for publication in highest-rating journal like Sci. Post Phys.
i. But I have one technical question concerning “a modest generalization of Thiele’s approach”. It seems to me, Eq. (27) contains a controversy: for 2D topological solitons, any form of \phi(r) has a singular point. The distribution of magnetization vector is free of singularities if and only if this singular point for \phi(r-r_2) coincide with the values of r_1, where theta = 0 or theta = pi. Thus, to be consistent, the ansatz (27) should include some complicated constrain that is not discussed here. At least, the coordinates r_1 and r_2 are not completely independent. Of course, the coefficients F_ij can be found at r_1 = r_2, as it is done in Eq. (31), but for general case r_1 \neq r_2 the above singularity manifest itself.
ii. On the best of my knowledge, the problem of the mass for 2D solitons, both topological and non-topological, was first considered in the article [B. A. Ivanov and V. A. Stephanovich, Two-dimensional soliton dynamics in ferromagnets, Phys. Letters A4, 89 (1989).]. In this article, no driving force was considered at all; and the soliton velocity was found to be the cause of deformation. The mass was found through the kinetic energy. How it can be compared with that in the referred article?
It seems to me, the answering of these two questions can be useful for potential readers. Concerning (i), some minor explanation can be done. The detailed discussion of the item (ii) is not mandatory, but could be very useful. After doing these changes, the article can be published in Sci. Post Phys.

  • validity: -
  • significance: -
  • originality: -
  • clarity: -
  • formatting: -
  • grammar: -

Author:  Oleg Tchernyshyov  on 2022-04-10  [id 2369]

(in reply to Report 2 on 2022-04-04)

Response to Anonymous Review 2. 1. The referee is right about the singularity in the field $\phi$. It exists somewhere in the middle of a skyrmion and can only be eliminated by having $\sin{\theta}=0$ at the same location. Therefore, the node of $\theta$ must, strictly speaking, follow the singularity in $\phi$. This consideration would seem to invalidate our assumption that $\theta$ and $\phi$ can have independent dynamics. Note, however, that in a skyrmion bubble $\sin{\theta}$ is very close to 0 away from the domain wall separating the two domains where $\cos{\theta}$ is close to $\pm 1$. Therefore, in practice, the dynamics of both $\theta$ and $\phi$ occur only in the vicinity of the domain wall and we need not worry about the singularity in $\phi$ occurring deep in the inner domain, where $\sin{\theta}$ is almost 0. In any event, the argument via independent fields $\theta$ and $\phi$ is (a) heuristic and (b) demonstrably works for a bubble domain. It would not be appropriate for a small skyrmion. Nonetheless, we think that our mechanical model with two massless particles would still work. 2. We have appended a discussion in the context of Ivanov and Stephanovich [28] to the end of the Conclusion. In a nutshell, the mechanism is the same (Doering's mass = kinetic energy traceable to a deformation). The crucial point is that the presence and the amount of deformation depend on the nature of the driving force. Adiabatic spin-transfer torque induces no deformation of a magnetic soliton and therefore does not create kinetic energy. 3. Aside from the points raised by the two referees, we have added a reference to Schuette, Iwasaki, Rosch, and Nagaosa [18], whose numerical work pointed to the possibility of massless dynamics from adiabatic spin torque.

---

## Round 4 · Referee Report · Anonymous (Referee 1) · 2022-4-25

Report

I think that the revised version can be published without additional corrections.

---

## Editorial Decision

published